# Mechanisms of Botulinum Toxin Type A Action on Pain

**DOI:** 10.3390/toxins11080459

**Published:** 2019-08-05

**Authors:** Ivica Matak, Kata Bölcskei, Lidija Bach-Rojecky, Zsuzsanna Helyes

**Affiliations:** 1Department of Pharmacology, University of Zagreb School of Medicine, Šalata 11, 10000 Zagreb, Croatia; 2Department of Pharmacology and Pharmacotherapy, Medical School, University of Pécs, Szigeti út 12, 7624 Pécs, Hungary; 3János Szentágothai Research Center, Center for Neuroscience, University of Pécs, Ifjúság útja 20, 7624 Pécs, Hungary; 4Department of Pharmacology, University of Zagreb Faculty of Pharmacy and Biochemistry, Domagojeva 2, 10000 Zagreb, Croatia

**Keywords:** botulinum toxin type A, pain therapy, migraine, neuropathic pain, mechanism of action

## Abstract

Already a well-established treatment for different autonomic and movement disorders, the use of botulinum toxin type A (BoNT/A) in pain conditions is now continuously expanding. Currently, the only approved use of BoNT/A in relation to pain is the treatment of chronic migraines. However, controlled clinical studies show promising results in neuropathic and other chronic pain disorders. In comparison with other conventional and non-conventional analgesic drugs, the greatest advantages of BoNT/A use are its sustained effect after a single application and its safety. Its efficacy in certain therapy-resistant pain conditions is of special importance. Novel results in recent years has led to a better understanding of its actions, although further experimental and clinical research is warranted. Here, we summarize the effects contributing to these advantageous properties of BoNT/A in pain therapy, specific actions along the nociceptive pathway, consequences of its central activities, the molecular mechanisms of actions in neurons, and general pharmacokinetic parameters.

## 1. Introduction

Botulinum toxin type A (BoNT/A) is among the most potent biological toxins in nature, and, together with serotypes B, E, and F, a cause of natural botulism in humans, which is characterized by flaccid paralysis of skeletal muscles and dysautonomia [1]. Based on early 19th century observations that a yet unknown toxin from contaminated food induced the symptoms of botulism, the idea emerged that a low dose of the agent could be used for overactive nerve disorder treatment. In the 20th century, different BoNT serotypes have been characterized and isolated, and Burgen et al. discovered that BoNT/A inhibits acetylcholine release from neuromuscular junctions (NMJs) in skeletal muscles [2]. These discoveries eventually lead to the development of its therapeutic use in the 1970s, when minuscule quantities of the toxin were first employed to correct the eye misalignment in strabism [3]. Today, botulinum toxin is the most commonly used therapeutic protein for the treatment of autonomic disorders, spasticity and hyperkinetic movement disorders, as well as in cosmesis for treating wrinkles [4]. BoNT/A is approved in upper limb spasticity, blepharospasm, hemifacial spasm, cervical dystonia, primary hyperhidrosis, and neurogenic detrusor overactivity [4,5].

BoNT/A effects on pain were reported in cervical dystonia in 1986 [6]. Initially, the analgesic effect in neuromuscular disorders and musculoskeletal pain was attributed to the muscle relaxant effect, until the antihyperalgesic effect in non-muscular pain models was unequivocally demonstrated in human patients and animal models [7]. A serendipitous discovery that BoNT/A injection into glabellar lines resulted in migraine resolution led to extensive investigation in headache, which demonstrated that BoNT/A treatment was beneficial in chronic, but not in episodic migraine [8,9]. Efficacy in chronic migraine was shown in two large industry-sponsored randomized controlled clinical trials (RCT), after which onabotulinumtoxin A (Botox^®^) eventually gained regulatory approval in 2011 [10]. So far, chronic migraine remains the only approved pain indication, even though several systematic reviews of clinical trials provide evidence for BoNT/A efficacy in different pain conditions [11]. Nevertheless, we have to add that, while the results are promising, the quality level of evidence is not yet high enough to provide explicit guidelines for pain physicians. Shortcomings of the available clinical research data include low number of participants, and thelack of standardized dosing and delivery protocols, as well.

In the present review, we summarize the available data on the mechanisms of action of BoNT/A on pain, considering both peripheral and central mechanisms along the nociceptive pathways. We also include an overview on the clinical evidence for chronic migraine and neuropathic pain syndromes. There is ample preclinical evidence that peripheral and central sensitization is effectively and safely alleviated by BoNT/A, but more clinical research is needed to determine the role of BoNT/A in chronic pain management.

## 2. Basic Pharmacology of BoNT/A: Mechanisms of Outstandingly High BoNT/A Potency and Long-Lasting Duration of Action

To describe how the effect of BoNT/A persists for months after its single use, it is important to understand the structure and function of the different toxin protein subunits, which mediate its high affinity recognition of neuronal targets and its intracellular enzymatic activity targeting the synaptosomal-associated protein 25 (SNAP-25), a part of heterotrimeric soluble N-ethylmaleimide sensitive factor attachment protein receptor (SNARE) complex (Section 2.1, Section 2.2 and Section 2.3). Apparently, the most important factor regulating the longevity of toxin action is the ability of BoNT/A protease to avoid cellular degradation mechanisms and survive in the cell cytoplasm for a long period (Section 2.4). In Table 1 we summed up the general factors affecting the BoNT/A potency and selectivity in relation to its pharmacological properties, as well as possible improvement of its use.

### 2.1. Structure of the BoNT/A Complex and Neurotoxin

From the lysed rod-shaped anaerobic bacterium *Clostridium botulinum*, BoNT/A is released as a large 900 kDa complex consisting of a 150 kDa neurotoxin and auxiliary proteins, which includes a non-toxic non-hemagglutinin (NTNH) and three hemagglutinin proteins. The NTNH part of the progenitor complex molecule contributes to toxin stability in the acidic environment and provides protection against proteases in the gastrointestinal system, while haemaglutinins are involved in BoNT/A translocation across the intestinal epithelial lining into the lymphatic system and the bloodstream [30]. Pharmaceutical formulations may consist of the entire complex (onabotulinumtoxinA), some of its components (500 kDa abobotulinumtoxinA), or the 150 kDa neurotoxin component only (incobotulinumtoxinA). It seems that the non-toxic components rapidly dissociate from the neurotoxin at the neutral pH within the injection site [31], and formulations containing the entire complex or only the neurotoxic component exhibit the same potency [32]. Thus, the only component of the complex that contributes to its efficacy upon injection seems to be the 150 kDa toxin [32].

The 150 kDa neurotoxin is a typical bacterial AB-structured toxin, consisting of heavy chain with membrane acceptor–binding and translocation domains, and a smaller 50 kDa light chain, a catalytic domain that mediates the intracytosolic proteolytic activity of the neurotoxin [33].

### 2.2. Pharmacokinetics of Injected BoNT/A

As mentioned, at the injection site under neutral pH, non-toxic components rapidly dissociate from the neurotoxin part of the molecule. The 150 kDa neurotoxin molecule, based on the injection technique and volume, spreads away from the injection site, which may induce side effects if non-targeted structures are reached. Examples include dysphagia in the treatment of cervical dystonia and paralysis of unwanted muscles after cosmetic use leading to ptosis. In pain therapy, BoNT/A is usually employed as a subcutaneous injection into multiple sites within the painful area, and intramuscular application into the multiple head and neck sites is approved for migraine treatment. Intradermal BoNT/A injection can also be used with the lower possibility for unwanted diffusion into nearby muscles compared to subcutaneous or intramuscular injection [34,35]. Perineural or nerve block treatments have also been suggested as effective modes of administration [25,26]. However, up to now, there have been no controlled studies that compare the analgesic potency or efficacy following different injection modes.

It is known that, after oral or inhalational exposure, the toxin can reach the systemic circulation by transcytotic transport across epithelial cells and further distributed to sensitive microcompartments, such as the proximity of cholinergic nerves endings. However, the systemic pharmacokinetics of BoNT/A used for therapeutic purposes (at doses not exceeding a few nanograms) has not been resolved, mostly because of too low toxin concentrations and the limitations of detection in circulating fluids. After local injections in tissues like the muscle, dermis, or subcutis, BoNT/A binds to the neuronal membrane and enters the neurons, while the unbound fraction is probably diluted in the lymphatic circulation and washed away from the injection sites, thus being unable to affect more distant neuronal endings because of too low concentration [33]. The time-course of BoNT/A entrance into peripheral neurons in vivo has been characterized and described only for larger, non-therapeutic doses, showing that BoNT/A might poison the peripheral motor terminals within minutes after its systemic injection [14]. Following radioiodinated BoNT/A injection into the muscle, the radioactivity was reduced to control levels within 12 h, suggesting that the toxin must enter the peripheral terminals within several hours to exert its paralytic activity before being diluted or degraded [12]. From the therapeutic point of view, it is very important to elucidate the distribution (a transport via axons and between cells) and degradation (pathways, enzymes, speed, and half-life) of both heavy and light chains. These could elucidate differences in duration of the BoNT/A action observed between species (human vs. mice/rat) and cell types (neuromuscular junction vs. autonomic cholinergic synapse vs. sensory nerves).

### 2.3. Specificity of BoNT/A Effect: Acceptor-Mediated Entrance into the Neuronal Cytosol

BoNT/A exerts high tropism for neurons based on its specific binding sites on neuronal terminals within the injected area. This specificity is due to a high affinity interaction of heavy chain and its neuronal binding sites. Heavy chain C-terminus (H_C_) binds to double acceptors consisting of gangliosides and synaptic vesicle 2 (SV2A-C) protein isoforms expressed at the extracellular side of the neuronal membrane. The ganglioside with higher affinity for BoNT/A, such as GD1a and GT1b, provide the surface for initial binding of BoNT/A (via heavy chain C-terminus) to neurons. Binding to SV2 protein enables the toxin’s further endocytotic entrance into endosome-like compartments, such as acidic synaptic vesicles (see detailed review by Pirazzini et al. [33]). The N-terminus of the heavy chain (H_N_) may also be involved into the specific neuronal binding via interaction with phosphatidyl inositol phosphates at the presynaptic plasma membrane [36].

It appears that the larger fraction of BoNT/A holotoxin undergoes sorting into acidic vesicles leading to prompt translocation into the cytosol, while the smaller fraction that enters non-acidic vesicular compartments may be sorted into the microtubule-dependent retrograde axonal transport pathways [37,38]. Currently, it is not known if the transport to non-acidic vesicles might be mediated by protein acceptors other than SV2. So far, BoNT/A has been demonstrated to enter the cell by binding to fibroblast growth factor receptor 3 [39], and possible interaction has been reported with the transient receptor potential vanilloid 1 (TRPV1) receptor activated by capsaicin and other vanilloid compounds, noxious heat, and lipid mediators [40]. The toxin light chain is released into the cytosol by energy- and pH-dependent pore-forming process involving thioredoxin thioreductase [33]. During the process, the light chain is separated from the heavy chain by reduction of the disulfide bridge and translocated through the transmembrane pore formed by the H_N_.

### 2.4. Longevity of Light Chain-Mediated Enzymatic Activity

The 50 kDa domain light chain is the catalytic domain that cleaves the SNAP-25 molecules. Naturally, any foreign protein in the cellular cytosol is the subject for proteasome-mediated degradation. BoNT serotype E (BoNT/E) is rapidly degraded in the cytosol with its light chain intracellular half-life being 1–2 days, leading to resolution of BoNT/E-mediated paralysis once the new SNAP-25 is synthesized. Unlike BoNT/E, which distributes evenly within the cytosol by diffusion, BoNT/A protease is concentrated at the inner side of the plasma membrane. It appears that double-leucine motif is the determinant of such BoNT/A localization. The binding of BoNT/A near the synaptic membrane involves interaction with septins, small GTP-ase proteins that polymerize into non-polar filaments to form a part of cytoskeleton [41]. Another possible explanation is that BoNT/A escapes the ubiquitine-proteasome degradation pathway by recruiting specialized enzymes that remove polyubiquitin chains [42]. The enzymatic activity of BoNT/A persists for up to one year in neuronal cultures, and up to five months in central neurons in vivo [43]. In comparison, the functional recovery of the NMJ and the duration of the antinociceptive activity of BoNT/A are shorter in humans (3–4 months), or even shorter in animals (around two weeks to one month). The duration of the neuromuscular paralysis depends on the dose, suggesting that higher dose leads to uptake of higher number of intracellularly active proteases in the cell [44]. In the sensory system, the effect of BoNT/A applied subcutaneously in the rat hind paw lasted for 15 to 25 days [45,46].

It was hypothesized that one of the major contributing factors to the long-term BoNT/A effect is the persistence of inactive SNARE heterotrimer in the presynaptic cleft [47]. However, after blockage of proteolytic activity by intracellularly delivered electroporated antibodies, the synaptic neurotransmitter release recovers after 4–5 days [48]. This time period is roughly similar to the period required for the turnover of synaptic SNAP-25, suggesting that turnover of SNAP-25(1–197) has a similar rate [49]. Thus, although SNAP-25(1–197) persistence within inactive SNARE heterotrimers might be contributing to the high potency of BoNT/A, the long-term efficacy is most likely affected by the persistence of the BoNT/A protease in the cell.

### 2.5. Inhibition of Neurotransmitter Release and the Effect on SNARE Supercomplex

Upon translocation into the cytosol, the light chain Zn^2+^-dependent metalloprotease enzymatically cleavs one of the conserved cleavage sites in the SNAP-25 polypeptide (SNARE motifs). BoNT/A and BoNT/E cleave the SNAP-25 at different SNARE motifs, forming truncated products of different lengths. The BoNT/A-truncated product termed SNAP-25 (1–197) is lacking only nine amino-acids at the C-terminus, while BoNT/E-cleaved product (SNAP-25(1–180) lacks 26 residues. Unlike BoNT/B and BoNT/E, whose efficacy is dependent on the disrupted formation of the SNARE heterotrimer consisting of SNAP-25, syntaxin, and VAMP-2/synaptobrevin, it appears that BoNT/A-mediated SNAP-25 cleavage does not affect the formation of the heterotrimer. Possibly due to preserved interaction with other SNAREs, SNAP-25(1–197) is not readily cleared away from the presynaptic terminal unlike BoNT/E-truncated product SNAP-25(1–180) [50]. In turn, SNARE-heterotrimer complex containing SNAP-25(1–197) competes with normal SNARE complexes at the vesicular release site, which is supported by finding that recombinant SNAP-25(1–197) inside the cell leads to neurotransmitter release blockade by producing a membrane-bound product [51]. This explains the disproportionate percent inhibition of synaptic neurotransmitter release evoked by BoNT/A in comparison to percentage of cleaved SNAP-25 molecules [49]. In vitro investigations suggested that only a small fraction (2–20%) of SNAP-25 molecules leads to complete synaptic blockade [52,53]. This implies that BoNT/A, by cleaving only a small subset of SNAP-25 molecules, blocks the transmitter release completely. Possibly, only 1–2 molecules of BoNT/A light-chain per synapse may be enough for the blockage of entire synaptic release machinery at individual synapses. Another factor in BoNT/A exquisite potency is the fact that the vesicle release site involves a formation of radial super-complex of SNARE heterotrimers, acting in concert to release one synaptic vesicle at a time. It appears that cleavage of only one or two molecules of SNAP-25 within the SNARE supercomplex disrupts the entire supercomplex function [54]. Thus, it is likely that two major factors affecting its potency are (1) specific binding of neuronal terminal and (2) disruption of small but vital population of synaptic SNAP-25 at the site of transmitter release.

### 2.6. Selectivity for Excitatory Synapses and Ca^2+^ Dynamics

Blockage of neurotransmitter release leads to build-up of synaptic vesicles near the presynaptic membrane. Apparently, BoNT/A is highly potent to block acetylcholine release and less potent to inhibit most other neurotransmitters such as glutamate, noradrenaline, serotonin, substance P, calcitonin gene-related peptide (CGRP), adenosine triphosphate (ATP), nicotinamide adenine dinucleotide (NAD), etc. [33,55,56], which might depend on the level of expression of high-affinity protein acceptors in the neuronal membrane.

SNARE-mediated vesicular release machinery requires a Ca^2+^-mediated signal to undergo conformational changes leading to vesicular membrane fusion with presynaptic plasma membrane. This is provided by interaction of vesicle-associated calcium sensor protein synaptotagmin with the C-terminal of SNAP-25 [57]. The main effect of BoNT/A-mediated SNAP-25 cleavage is a reduced affinity of intracellular Ca^2+^ sensor synaptotagmin to SNAP-25. Under normal, physiological concentration of Ca^2+^, the lack of 9 amino-acids at the C terminus of SNAP-25(1–197) impairs this interaction. However, the interaction is restored under high Ca^2+^ concentration, and SNARE complex regains its normal neurosecretory function [58]. Thus, any treatment aimed at increasing the Ca^2+^ concentration restores the neurosecretory function of BoNT/A-poisoned synapse, which may be responsible for observed lack of BoNT/A effect on capsaicin-evoked release of peptides in few in vitro studies [59,60] (see ref. [61]).

Regarding the in vitro release of inhibitory neurotransmitters, in the mouse embryonic spinal cord, BoNT/A appears to block the evoked release of glycine at similar potency compared to ACh. However, it appears that BoNT/A does not block the release of GABA in adult neurons. It has been posited that inhibitory neurons do not express SNAP-25, and that the neurotransmitter release in GABA-ergic neurons might be mediated by some other SNAP-25 isoform (e.g., SNAP-23) [62]. However, GABA-ergic neurotransmitter release is also invariably modulated by SNAP-25, since gene deletion of SNAP-25 expression abolishes the neurotransmitter release in GABA-ergic neurons. It appears that the effect of SNAP-25 cleavage is readily overcome by transient synaptic increase of intracellular calcium in adult inhibitory neurons. Interestingly, the level of SNAP-25 expression appears to be contributing to the calcium levels, since it interacts with ion channels as a negative regulator on voltage-gated calcium channels [63]. Artificial expression of high levels of SNAP-25, or application of Ca^2+^ chelators, confers GABA-ergic neurons more sensitive to BoNT/A. Thus, the level of SNAP-25 contributes to the low sensitivity of inhibitory neurons and, conversely, higher sensitivity of glutamatergic synapses to BoNT/A due to additional function of SNAP-25, which alters the calcium dynamics via its interaction with Ca^2+^ channels [64].

### 2.7. Interaction with Ion Channels and Pain-Sensing Receptor Translocation

By inducing cleavage of SNAP-25, BoNT/A may interfere with protein translocation from endosomal compartment to the cell plasma membrane. This has been proven for the TRPV1 capsaicin receptor, which is an important non-selective cation channel in pain transmission [65,66]. BoNT/A might interfere with the function of Na^+^ channels as well [67], possibly also due to this mechanism. Based on investigation of mechanical sensibility of dural afferents, it was proposed that BoNT/A may decrease the activity of mechanosensitive receptors and the transient receptor potential ankyrin 1 (TRPA1) channels [68,69] (their molecular identity not being characterized in mentioned studies). These mechanisms have been proposed to contribute to the BoNT/A antinociceptive activity.

## 3. BoNT/A Effects on Peripheral Sensory Nerves

### 3.1. Prevention of Nociceptive Neurotransmitter Release in Peripheral Terminals

While it was initially believed that BoNT/A antinociceptive effects are mediated by its actions on the muscles, the findings that a broad range of pain conditions not related to muscular contraction were also relieved by BoNT/A have soon suggested its possible effects on the sensory neurons. In analogy with its known action on the neuromuscular junction, it was proposed that BoNT/A prevents the sensory neurotransmitter release from peripheral sensory nerve endings. There is a plethora of studies that demonstrated the blockade of nociceptive neurotransmitter release by BoNT/A in vitro from peripheral sensory nerves. In primary sensory neuronal cultures, BoNT/A blocks the KCl-evoked release of CGRP and substance P [61,70], suggesting that sensory neuropeptide release is dependent on the SNARE complex. This has been confirmed in ex vivo urinary bladder preparations [71,72]. Studies involving formalin-induced stimulation of rat hind-paw and temporomandibular joint reported a reduced elevation of tissue content of glutamate and substance P by peripherally injected BoNT/A [7,73]. In human skin, intradermally injected BoNT/A reduces the capsaicin- and heat-evoked glutamate release measured by microdialysis [74]. Although BoNT/A prevents peripheral nociceptive transmitter release, preclinical data provided no evidence that such peripheral toxin action is causally involved in its antinociceptive effect. Moreover, it was shown that BoNT/A antinociceptive action is not causally related to toxin’s anti-inflammmatory effects, which are presumably mediated by prevention of peripheral neurotransmitter release, either (Section 3.2). In addition, antinociceptive effect of BoNT/A was demonstrated in centrally-mediated pain models (Section 5).

### 3.2. Anti-Inflammatory Effects of BoNT/A

In contrast to consistent evidence for BoNT/A inhibitory action on pain of different etiologies, BoNT/A’s effect on inflammation is still inconclusive, mostly because of contradictory animal experimental results. Cui et al. [7] were the first to show that intraplantar BoNT/A injection reduced formalin-induced edema and accompanied peripheral glutamate release, thus, suggesting inhibition of peripheral sensitization as the primary mechanism of BoNT/A action on pain and inflammation. In contrast, in acute inflammatory pain models evoked by intraplantar injection of carrageenan or capsaicin, BoNT/A pretreatment reduced pain hypersensitivity but failed to affect either carrageenan-induced paw edema or capsaicin-induced plasma protein extravasation both at macroscopic and histological levels [75]. The observed dissociation between the effects on pain and local inflammation questioned the well-established concept about the common peripheral mechanism of BoNT/A action. In models of cyclophosphamide-induced cystitis and capsaicin-induced prostatitis, local administration of BoNT/A decreased bladder hypersensitivity and neurogenic inflammation (measured as decreased concentrations of CGRP and SP) as well as inflammatory cell accumulation and cyclooxygenase-2 expression in the prostate and spinal cord [76,77]. Furthermore, the anti-inflammatory action of BoNT/A was tested in models of acute or chronic arthritis. Reduction of long-lasting complete Freund’s adjuvant (CFA) - induced joint inflammation and destruction shown by decreased inflammatory cell infiltration around the articular cartilage and synovial membrane were observed for two weeks after intraarticular BoNT-A application [78]. Additionally, intraarticular BoNT/A injection decreased CFA-induced expression of the proinflammatory cytokines IL-1β or TNF-α in the synovial tissue, accompanied by alleviation of cartilage degeneration and inflammatory cell infiltration [79]. Intraarticular BoNT-A reduced the persistent inflammatory hypersensitivity induced by systemic CFA and intraarticular methylated BSA in the temporomandibular joint as well, and significantly diminished the peripheral release of the SP, CGRP, and IL-1β [73].

Human data about the peripheral anti-inflamatory effects are also inconsistent. Bittencourt da Silva et al. found a reduction of glutamate release in dermal microdialysates by BoNT/A injections in healthy volunteers after capsaicin and thermal provocation [74]. In contrast, Attal et al. [45] in skin punch biopsy specimens of patients with peripheral neuropathic pain found no difference in SP and CGRP content between the BoNT/A and the saline-treated control group (however, in the mentioned study, CGRP and SP content was not compared to normal healthy controls). While the first study proposed the involvement of transmitter release inhibition in the analgesic action of BoNT/A, the second study questioned the role of peripheral neuropeptides in the toxin’s analgesic effect, at least in peripheral neuropathic pain.

Important insight into the mechanism of antinociceptive action of BoNT/A came from a set of experiments on neuropathic pain models, whose main findings are shown in Table 2.

### 3.3. Involvement of BoNT/A Systemic Effect in the Measurement of Nociceptive Responses

In the majority of animal studies described above, the antinociceptive effect of the toxin has been studied by measuring motor responses, such as hind-paw withdrawal following non-painful or painful mechanical or thermal stimuli. Generally, when assessing the analgesic efficacy of substances affecting motor performance, the effect on the motor performance of the drug itself might confound the results of nociceptive pain measurement. Due to possible BoNT/A diffusion from the site of toxin injection into the bloodstream, especially at higher doses, some systemic effects might be expected to result in decreased overall motor performance, and thus, possibly affecting the pain-evoked motor responses. Interestingly, within low dose BoNT/A range in rats (3.5 U/kg–15 U/kg), it has been observed that the pain-evoked motor response usually peaks with the lowest effective dose [7,45,75]. However, some animal studies reported dose-response effect by taking into account high doses which most likely affected the motor performance. In rats, Cui et al. [7] found reduction of formalin-evoked phase II spontaneous response at 30 U/kg compared to 3.5–15 U/kg doses, and Park et al. [80] also reported reduction of allodynia based on comparison of 10 U/kg, 20 U/kg, 30 U/kg, and 40 U/kg doses. Thus, in studies involving BoNT/A, possible systemic effect of employed BoNT/A dose has to be ruled out first by testing the motor performance after particular mode of toxin injection.

### 3.4. Regenerative Effects of BoNT/A in the Injured Nerve

An interesting set of observations from neuropathic pain model based on chronic constriction injury (CCI) suggested that BoNT/A affects the functional recovery of injured peripheral nerves. BoNT/A injected intraplantarly in neuropathic mice improved the sciatic index and weight bearing, along with increased cell division cycle 2 (cdc2) protein expression and Schwann cell proliferation and maturation [84,85]. Further study indicated that BoNT/A might be axonally transported within the sciatic nerve trunk and enter the Schwann cells [96]. Interestingly, BoNT serotype B (BoNT/B), which also counteracts the pain evoked by CCI, does not possess the nerve regenerative ability [97]. This suggests an additional role of cleaved SNAP-25 or some yet unknown effects of BoNT/A on gene expression patterns within the injured nerve, which are not affected by BoNT/B.

### 3.5. Effects of BoNT/A on the Sensory Ganglia

In the sensory ganglia of injured nerves, BoNT/A reduces the pain-evoked upregulated protein expression of nociception-related ion channels such as TRPV1, purinoceptor P2X3, and reduces the mRNA expression of pronociceptive peptides such as preprodynorphin [46,85,87]. Based on reduced surface expression of the TRPV1 receptor protein, but not its mRNA, it was proposed that BoNT/A might block the translocation of TRPV1 to the sensory neuronal surface in the ganglia. This was corroborated by in vitro studies of primary sensory neuronal cultures [65,66]. In trigeminal ganglion primary sensory neurons acutely isolated from animals with infraorbital nerve constriction (IoNC) and BoNT/A or saline injection into the whisker pad, BoNT/A prevented the KCl-evoked neuroexocytosis [98]. Shimizu et al. [66] demonstrated that BoNT/A reduced the TRPV1 expression in sensory neurons projecting from the dura mater. It was reported that BoNT/A might be axonally transported from the periphery to dural CGRP-expressing primary afferents [22]. Since peripheral orofacial area and dura are not innervated by the same sensory neurons, these findings indicate BoNT/A trans-synaptic transport between sensory neurons that innervate different intracranial and extracranial targets [22,99]. This mechanism of BoNT/A traffic provides possible explanation for BoNT/A efficacy in migraine pain associated with meningeal afferents. It was also hypothesized that BoNT/A, if transported into satellite glial cells, might modulate the release of glutamate from glial cells interacting with sensory neurons and alter the intraganglionic communication between the glia and sensory neurons [100]. However, this possibility has not been examined in vivo.

BoNT/A injected directly into the rat trigeminal ganglion reduced the orofacial formalin-induced pain [101], and infraorbital nerve constriction (IoNC)-induced trigeminal neuropathic pain [86]. Although the ganglia might be an important site of toxin action contributing to its antinociceptive activity, it seems that BoNT/A action within the ganglia is not sufficient. Injection of colchicine into the ganglion before the BoNT/A intraganglionic injection prevented the toxin effect on pain. In addition, if the site of action was the ganglion, BoNT/A i.g. injection should have induced a fast antinociceptive effect visible after 24 h. However, the analgesic action of such injection was delayed—it was evident after two days. Thus, BoNT/A action on pain is dependent on axonal transport beyond the ganglion, most likely into the CNS [86,101].

## 4. Actions of BoNT/A in the Central Nervous System

The antinociceptive effect of BoNT/A was investigated in several pain models wherein the unilateral tissue injury induces a long-lasting bilateral pain hypersensitivity. In these models, it has been accepted that pain development, its spread to contralateral side, and chronification is most probably mediated by complex spinal and supraspinal mechanisms [102,103]. In the model of intramuscular (i.m.) acidic saline induced “mirror pain,” BoNT/A injected into the ipsilateral hind paw pad significantly reduces mechanical hypersensitivity on the injured but also on the uninjured contralateral side [104]. Bilateral effect of BoNT/A was repeatedly shown in other bilateral pain models, like in trigeminal neuropathy induced by unilateral infraorbital nerve constriction injury (IoNC) [86], inflammatory pain induced by complete Freund’s adjuvant (CFA) injection into temporomandibular joint [22], and hyperalgesia after carrageenan injection into the calf muscle [24]. The toxin’s bilateral effect after unilateral injection was demonstrated in poly-neuropathic states evoked by systemic paclitaxel [82] or streptozotocin [83], as well as in the model of bilateral acute model of s.c. carrageenan-induced inflammatory hyperalgesia [94].

When injected into the spinal canal, BoNT/A relieved pain faster and in lower doses (within 24 h, 1–2 U/kg) compared to local subcutaneous (s.c.) injection (3–5 days; 3.5–7 U/kg). However, it was completely ineffective if applied supraspinally, e.g., to cisterna magna [24]. The effect on bilateral pain depends on the axonal transport, since the axonal transport blocker colchicine prevented the reduction of pain on both sides. These consistent behavioral observations strongly support the BoNT/A central effect at the level of spinal cord segment associated with peripherally innervated area. Immunohistochemical studies of BoNT/A-cleaved SNAP-25 demonstrated that BoNT/A is axonally transported into the sensory regions of brainstem or spinal segment associated with the peripherally injected area. Following facial injection, BoNT/A-cleaved SNAP-25 was visible only in pain-associated areas of the spinal trigeminal nucleus system (trigeminal nuclei oralis and caudalis [24,105].

Interestingly, BoNT/A application on the side contralateral to the injuries evoked by acidic saline or carrageenan did not lead to bilateral action. In the model of acidic saline-induced chronic mechanical hypersensitivity, contralateral BoNT/A reduced the pain only on the injection side [104]. On the other hand, in carrageenan-induced mechanical hyperalgesia model, contralaterally injected BoNT/A failed to affect pain on either side [24]. Based on these results, we may speculate that the bilateral effect of BoNT/A depends on the type and mechanisms of the injury (intensity and substance used to provoke tissue damage, time-course of bilateral pain development, etc). Observations of unilateral activity or no activity after contralateral BoNT/A injection suggest that BoNT/A is not necessarily transported to the contralateral side.

### 4.1. Effects in TRPV1 Receptor-Expressing Central Afferent Terminals

BoNT/A does not affect any other sensory sensation, apart from pain-related inflammatory and mechanical stimulation. One of the possible explanations for this selectivity, in contrast to other sensory modalities, is the selective BoNT/A entrance into particular sensory neuron population (Figure 1). It was shown that the effects of BoNT/A on orofacial formalin-induced pain were prevented by the destruction of TRPV1-expressing afferents evoked by high dose capsaicin injection into the trigeminal ganglion [21]. In addition, the SNAP-25 cleavage in the trigeminal nucleus caudalis was also abolished and prevented by trigeminal denervation with capsaicin. This study suggested the occurrence of BoNT/A enzymatic activity in central afferent terminals of capsaicin-sensitive TRPV1-expressing neurons (Figure 1). The exact reason for this selectivity is not known, but the capsaicin-sensitive primary afferents might be more prone to acceptor-mediated BoNT/A entrance at the periphery. It is also possible that sensory neurons in neuropathic or other pain conditions are expressing higher levels of SV2A proteins, which might facilitate the BoNT/A entry only into the sensitized sensory neurons [106]. In line with necessary role of sensory neurons, patients with more allodynia or better preserved sensory function better respond to the pain treatment with BoNT/A [34,107]. Moreover, the importance of capsaicin-sensitive neurons is supported by the observation that neuropathic patients with lower thermal deficits responded better to BoNT/A treatment [107]. In different homozygous knockout mice with gene deletions of neurotransmitters and receptors related to capsaicin-sensitive neurons (TRPV1, NK1 receptor, and substance P/neurokininA), BoNT/A failed to reduce the neuropathic and inflammatory pain [21,95]. These findings suggest important role of functional transmission within capsaicin-sensitive neurons for exertion of BoNT/A antinociceptive activity. However, in bilateral carrageenan-induced chronic mechanical hyperalgesia, the desensitization of sciatic nerve with inrasciatic high-dose capsaicin treatment did not affect BoNT/A’s bilateral antinociceptive effect [24], nor did capsaicin itself affect the contralateral carrageenan-evoked pain [24]. Thus, it may be speculated that, at least in sciatic innervation area, BoNT/A might also require other types of neurons to achieve this bilateral effect.

### 4.2. Indirect Central Actions on the Endogenous Opioid and GABA Neurotransmission

Drinovac et al. [27,89], based on several lines of experiments on rats with pain in the sciatic region, suggested that enhancement of the opioid and GABA neurotransmission mediated by their receptors (µ-opioid and GABA-A) is involved in the central antinociceptive effect of BoNT/A. Although the opioid receptor antagonist naltrexone, as well as the GABA-A antagonist bicuculline, reduced BoNT/A effect on pain when applied intraperitoneally or intrathecally, the antagonistic effect was completely absent after their intracisternal and intracerebroventricular injection, thus providing evidence for segmental intraspinal action of BoNT/A on pain [24]. Furthermore, the analgesic action in the trigeminal innervation region also involves interactions with the central, not peripheral, endogenous opioid system, most likely at the level of trigeminal nucleus caudalis [108]. Interestingly, in the CCI-induced neuropathic pain model of mice, BoNT/A co-administration increased the morphine-induced analgesic response and reduced the tolerance to repeated morphine application accompanied with enhanced neuronal µ-opioid receptor expression [88,109]. It thus seems that modulatory spinal inhibitory neurotransmitter system, which is known to attenuate sensory input to the dorsal horn, plays a significant role in the central antinociceptive action of BoNT/A. However, the mechanism of this interaction is still not clear. It possibly involves some yet unidentified neuronal circuits within the spinal cord rather than direct action of BoNT/A on the inhibitory neurons. Opioid and GABA and transmissions have a role in the attenuation of sensory input to the spinal dorsal horn. Similarly to BoNT/A, different treatments suppress the morphine-induced pain tolerance, such as chronic N-methyl-D-aspartate receptor (NMDAR), neurokinin 1 receptor (NK1), or TRPV1 antagonist treatments and desensitization of capsaicin-sensitive neurons, etc. [110,111,112]. GABA receptor positive allosteric modulators, such as ethanol and diazepam, also suppress morphine-induced tolerance, and their effects are reversible by bicuculline [113]. It can be hypothesized that BoNT/A affects a common pathway inducing morphine tolerance, e.g., synaptic transmission via glutamate from primary afferent terminals to the secondary sensory neurons in the spinal dorsal horn. This might consequently induce a compensatory increase in the morphine-suppressed endogenous opioidergic inhibitory neurotransmission. BoNT/A was shown to be effective in headache patients resistant to analgesics and having medication overuse [28,29], which is in line with its ability to restore the normal opioidergic transmission in the trigeminal nucleus caudalis and spinal dorsal horn.

### 4.3. Effects on Astroglia and Microglia (Neuroinflammation)

Activated microglia and astrocytes have a well-known role in the progression and maintenance of neuropathic pain [85]. Hence, a potential interference of BoNT/A with glial cells was investigated in several neuropathic pain models in mice and rats (Table 2). Mika et al. [85] showed that intraplantar BoNT/A injection reduced CCI-induced mechanical and thermal hypersensitivity and microglia (but not astrocyte) activation in the ipsilateral lumbar spinal cords in rats. Attenuation of the microglia activation and neuroinflammation was proposed to play a role in the overall antinociceptive action of BoNT/A. Furthermore, in the same model in mice, it was demonstrated that intraplantar BoNT/A injection reduced microglia activation and astrocyte activiation in both the dorsal and ventral horns of the spinal cord [88]. In CCI-exposed rats Zychowska et al. [93] showed that BoNT/A diminished microglia activation, the levels of the pro-inflammatory citokines IL-1β and IL-18, and enhanced the concentrations of the inhibitory interleukins IL-1RA and IL-10 in the spinal cord and/or the DRG, suggesting BoNT/A-mediated reinstatement of neuroimmune balance deteriorated by the nerve injury.

Those in vivo experiments drew attention to the involvement of glia in the antinociceptive effect of BoNT/A, at least in neuropathic pain models, however, nothing was known about the nature of this interaction. A recent in vitro study demonstrated that BoNT/A inhibited the expression of pro-inflammatory IL-1β, IL-18, IL-6, and nitric oxide synthase 2 (NOS2) through the inhibition of p38-, ERK1/2-, and NF-κB-mediated intracellular signaling pathways on primary rat microglia, but not astrocyte cell line after lipopolysaccharide stimulation. Additionally, it decreased the expression of SNAP-23 (the main SNAP molecule in microglia) and increased TLR2 expression, which is suggested to be its microglial molecular target [114]. These results are in line with previous observations on a murine macrophage cell line, where BoNT/A induced changes of global gene expression through a TLR2-dependent pathway [115] These results propose a direct interaction of BoNT/A with microglia cells and reveal interference with some intracellular signaling processes as an explanation for in vivo findings. This topic needs further investigations to reveal the details of BoNT/A action on the neuroinflammatory mechanisms.

### 4.4. Effects on the Ascending Pain Processing Pathway

As mentioned, the main site of the antinociceptive effect of BoNT/A is situated at the level of segmental spinal dorsal horn and/or the brainstem sensory region, associated with the toxin-injected area. Studies in the optic nervous system demonstrated the toxin’s sequential axonal transport and transcytosis over several synapses, similarly to tetanus toxin [116,117]. Thus, it is possible that BoNT/A, following transcytosis in the medullary or spinal dorsal horn, might be transported into the ascending sensory regions by a similar mechanism.

Up to now, this has been examined by Matak et al. [21]. After BoNT/A injection into the rat whisker pad, cleaved SNAP-25 was found only in the spinal trigeminal nucleus caudalis and oralis, but not in other sensory regions examined (thalamus, hypothalamus, sensory cortex, locus coeruleus, periaqueductal gray, etc.). This study suggests that the spread of BoNT/A within the CNS is relatively small and the toxin transport to more distant sensory and motor regions is unlikely at small doses applied. However, high BoNT/A doses might lead to the occurrence of BoNT/A-cleaved SNAP-25 in contralateral sensory and motor regions, revealing a trans-synaptic traffic within commissural spinal cord neurons [118].

An indirect effect of the toxin on neuronal activation in distant sensory regions has also been described [21]. Orofacial formalin injection-evoked c-Fos expression was suppressed by BoNT/A in the periaqueductal gray and locus coeruleus. The toxin did not alter c-Fos expression in other sensory regions in the diencephalon related to the motivational and affective pain modalities, such as the hypothalamus, paraventricular thalamic nucleus, and amygdala. This observation suggests an indirect modulation of neuronal activation in the ascending sensory regions. However, BoNT/A might have a smaller effect on other pain modalities related to motivational and affective pain processing.

In the orofacial formalin test, BoNT/A injected into the whisker pad did not modulate the levels of monoamines and their metabolites, not supporting their in the BoNT/A antinociceptive action [119]. BoNT/A’s effect on ascending pain processing in humans has not been examined by functional imaging studies. However, a recent study in neurogenic overactive bladder indicated effects of BoNT/A on most brain sensory regions known to be involved in the sensation and process of urinary urgency, thus indicating BoNT/A effects beyond the bladder [120].

## 5. An Overview of Clinical Evidence of BoNT/A Analgesic Efficacy

As we previously discussed, the analgesic effect of BoNT/A has been shown in the treatment of chronic migraine, including medication-overuse headache and several localized neuropathic pain syndromes—all of them pain conditions that cannot be adequately alleviated with conventional analgesics. The pharmacotherapeutic options for the treatment of chronic migraine and neuropathic pain are very limited, with certain antiepileptics and antidepressants [121,122], and the development of analgesics with novel mechanisms of action has proven to be extremely challenging [123]. Since the pathophysiology of pain in these syndromes is complex, involving the sensitization of primary and secondary afferents and also the enhanced activation of glial cells and reduced activity of the endogenous pain modulating mechanisms, the peculiar long-lasting actions of BoNT/A on multiple levels of the pain transmission as described in detail above offer a unique treatment modality. In comparison to other classical and non-classical analgesic drugs, the greatest advantage of BoNT/A use is a sustained effect after a single application and low risk for adverse effects even upon repeated administrations.

While observational clinical data suggested a potential analgesic effect and in vivo animal experiments consistently demonstrated an anti-hyperalgesic effect in a variety of models (Table 2), the first experimental human studies did not demonstrate an unequivocal analgesic effect. In initial studies on healthy human volunteers, the pain thresholds to heat, cold, or electrical stimulation as well as capsaicin-induced pain ratings were unchanged after subcutaneous or intradermal application of BoNT/A [124,125,126]. While baseline heat and cold pain thresholds were consistently unaltered in later studies as well, it was demonstrated that BoNT/A increased mechanical pain thresholds [69] and significantly alleviated capsaicin-induced pain and allodynia [127,128], mustard oil–induced pain, and histamine-induced itch [69]. In all cases, a significant reduction in the neurogenic vasodilation was also measured, suggesting a reduced release of neuropeptides from capsaicin-sensitive nerve terminals. Importantly, the effect of BoNT/A on intradermally applied capsaicin-induced responses was demonstrated both by intramuscular and subcutaneous delivery [127,128].

To date, the highest level of clinical evidence exists for the efficacy of BoNT/A in chronic migraine supported by several RCTs and metaanalysis [8,9]. The approved delivery route is intramuscular, and the dosing is 5 U per site at 31–39 precise anatomical locations of the head and neck at 12-week intervals. It was the first and, up to now, it remains the only approved clinical use of BoNT/A for a pain condition. Initial open-label studies also suggested that BoNT/A might be effective to reduce headache severity in other primary headache types, but later data from RCTs could not confirm a significant reduction of headache frequency or severity in either episodic migraines or tension-type headaches. The lack of a more general efficacy in headaches is not clear yet, since BoNT/A could reverse the sensitization of meningeal nociceptors in rats [68] and it was clearly effective in experimentally-induced acute trigeminal pain in humans [127,128]. A recent Cochrane metaanalysis of 4 RCTs conducted for chronic migraine found that the overall reduction of migraine days compared to placebo was -3.07 days. This could be considered as a modest improvement of symptoms, but one has to appreciate that the safety profile of BoNT/A was excellent based on data from 23 RCTs. There were few adverse effects, among which the most relevant were ptosis, muscle weakness, and neck pain, which were mostly transient. Discontinuation rates were very low, which also suggests an overall good tolerability. A postmarketing observational study which followed patients for 108 weeks also strongly supports the long-term safety of repeated BoNT/A treatments [20]. This is especially relevant since adherence to oral preventive antimigraine drugs is estimated to be extremely low [129,130].

However, controversy also exists about the efficacy of BoNT/A in chronic migraine treatment. A recent double-blind randomized controlled trial on 90 patients suffering from chronic migraine with medication overuse did not show any benefit from BoNT/A (155 IU) in addition to medication withdrawal regarding the reduction of headache days and improvement of patients’ quality of life in comparison to the placebo group. This trial is methodologically different from most of the previous studies on chronic migraine because the placebo-group was injected with the masking doses of BoNT/A (17.5 U) in the forehead to prevent facial wrinkling. The authors assume that the unblinding in previous experiments likely could positively affect the modest therapeutic benefit of BoNT/A. [131].

Attempts were also made to identify predicting factors associated with a better response to BoNT/A treatment. Jakubowski et al. established that the imploding and ocular headache was more frequently reported by responders, while the majority of non-responders described their headache as exploding [132,133]. The authors theorized that the difference in headache character could be derived from the sensitization of a different population of nociceptors—extracranial in the case of responders, and intracranial in the case of non-responders. Another group found that the plasma level of CGRP was higher in responders compared to non-responders, and a decrease of CGRP by treatment could also be demonstrated [134,135]. On the other hand, the presence of cutaneous allodynia did not prove to be a predicting factor for the efficacy of BoNT/A, even though it is considered to reflect the presence of central sensitization [132,133,136]. A recent study on Korean patients also suggested that longer disease duration could be associated with a poorer response [137].

RCTs also showed significant analgesic efficacy of BoNT/A in several neuropathic pain conditions, such as trigeminal, posttraumatic or postherpetic neuralgia [11,138,139]. The onset of clinical efficacy could be detected at week 1 in most reported trials. The duration of effect of a single administration was followed up to three months. Heat and cold sensation was not affected by neuropathic pain patients either [107], confirming the selective action of BoNT/A. The evidence is convincing, especially the remarkable efficacy in trigeminal neuralgia, although we have to add that the number of patients included in these trials and the placebo responses were small. Data from a larger industry-sponsored trial for postherpetic neuralgia did not show significant reduction of pain scores [140]. The most relevant RCTs on neuropathic pain are compiled in Table 3.

Small but significant pain relief was demonstrated in RCTs for musculoskeletal pain, in particular plantar fasciitis, tennis elbow, and low back pain. The efficacy could not be conclusively proven in patients with myofascial pain syndromes [11,148]. In the case of osteoarthritis, some trials have demonstrated efficacy for patients with refractory osteoarthritic pain. Results for RCTs for osteoarthritic pain are summarized in Table 4.

Because of the lack of clear guidelines regarding the amount of BoNT/A injected, the route and site of injection, and the number of injections, clinical studies vary in treatment protocols, which make them difficult to compare. Thus, future clinical trials should be more standardized regarding study design, patients’ stratification, measurement outcomes, and study endpoints, and those studies should look for both short- and long-term outcomes relevant for BoNT/A efficacy, tolerability, and safety.

## 6. Conclusions

The success of BoNT/A therapy in treating certain chronic pain conditions and individual patients may be influenced by a variety of factors related to its main mechanisms of action. In the sensory system, BoNT/A action may be restricted to certain neuronal populations mediating pain hypersensitivity, which could explain its efficacy only in some types of chronic pain or patient subpopulations. Longevity of action of BoNT/A is most likely to be related to its cellular localization, which enables a long-term effect after a single application. Further characterization of BoNT/A effects on multiple sites of action on its way from the periphery to CNS are the next necessary steps to explain its antinociceptive effect and help to improve its clinical use.

## Figures and Tables

**Figure 1 toxins-11-00459-f001:**
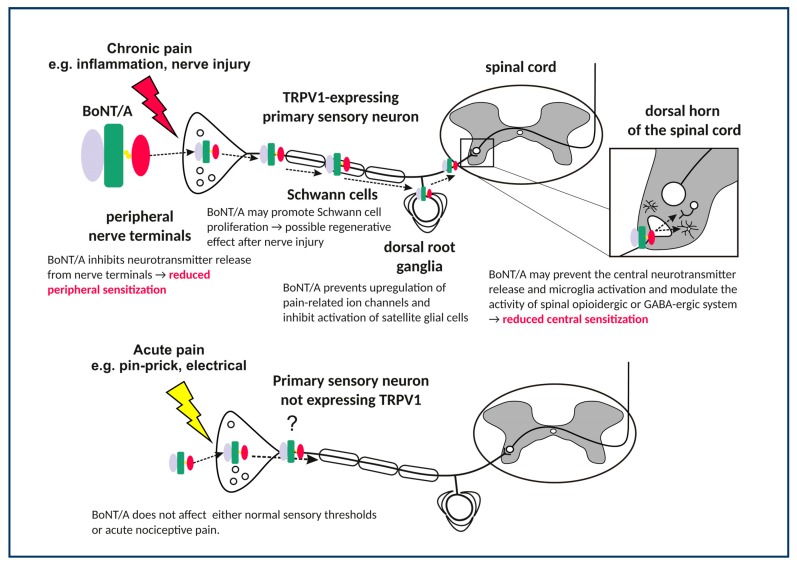
Actions of BoNT/A along the pain pathway.

**Table 1 toxins-11-00459-t001:** General and pain-specific factors influencing the properties and peculiarities of botulinum toxin type A (BoNT/A) action.

General Factors not Specific to Pain				
Property of BoNT/A Molecule or Peculiarity of Action	Mechanisms of Action	Contribution to Desirable Pharmacological Properties	Attempted or Potential Improvement	References
Low local diffusion after application	rapid and high affinity binding to neuronal membrane at the injected site	safety, onset of the effect	lower injection volume, intradermal injections	[12,13,14]
Absorption through epithelial barriers	crossing epithelial barriers by transcytosis	application by different routes (e.g., transmucosal)	novel therapeutic systems with incorporated BoNT/A which might improve toxin absorption and extend the contact time with the epithelial tissue/mucosa	[15]
Specificity for hyperactive neurons	Expression of membrane acceptors such as glycosylated SV2C; higher rate of SVs exo/endocytosis favors toxin uptake	safety, selectivity for hyperactive nerve terminals	recombinant chimeras with different neuronal specificities	[16]
Protease specific targeting of SNARE proteins	synaptic localization, disturbance of SNARE supercomplex	potency, safety	- specific point mutations for higher affinity to SNAP-25	[17]
			- recombinant molecules with shorter action or different affinity for SNARE proteins	
Protease longevity inside neurons	cellular localization, avoidance of proteasomal degradation	long duration of effects	specific chimeras that change the affinity for intraneuronal degradation system	[18]
Reversibility of the neuroparalysis	recovery of neuronal exocytosis is dependent on nerve terminal type (gain of function)	long duration of effect (from 3 months to more than a year)	interference with the nerve function recovery processes	[19]
Repeatability of neuroparalysis	recovery of neuronal exocytosis can be repeated many times without loss of neuron function	application schedule	repeated application for prolonged period into the same site	[20]
**Factors Specifically Related to Pain**				
**Property of BoNT/A Molecule or Peculiarity of Action**	**Mechanisms of Action**	**Contribution to Desirable Pharmacological Properties**	**Attempted or Potential Improvement**	**References**
Selectivity for certain sensory neuron populations	occurrence in TRPV1-expressing neurons	selectivity for chronic or prolonged pain	recombinant chymeras with different receptor specificities	[21,22,23]
	effect on glutamatergic transmission	efficacy in chronic pain (possibly LTP related)		
	effect on peptidergic transmitters	efficacy in chronic pain and migraine		
Segmental activity in the sensory nucleus/spinal cord dorsal horn	microtubule-dependent neuronal axonal transport	localization of toxin effect	neural block for segmental treatment	[24,25,26]
Interaction with other pain neurotransmitter system	Interaction with endogenous opioid system	synergism with opioid analgesic and avoidance of tolerance development; efficacy in opioid-overused patients	combined use of lower dose opioids and BoNT/A	[27,28,29]
	Restoration of sensitivity to morphine			

**Table 2 toxins-11-00459-t002:** Key findings from neuropathic pain models.

Model	BoNT (Type; Dose, Application)	Findings/Comments	Ref.
partial sciatic nerve injury in rats	A; 7 U/kg; i.pl.; injected after established hypersensitivity (day 14)	Long-term reduction of mechanical and thermal hypersensitivity (from day 5 after injection). First study on experimental peripheral neuropathic pain.	[45]
ligation of L5/L6 spinal nerve in rats	A; 10, 20, 30 or 40 U/kg i.pl. after established hypersenitivity	Reduction of mechanical allodynia (after 1 day) and cold allodynia (three days after injection; both effects lasted for 15 days after injection). The effect was dose-dependent. However, large systemic doses were used.	[80]
chronic constriction injury of the sciatic nerve in mice	A; 15 pg/mouse; i.pl. pre- and post-injury	Reduced mechanical allodynia (from day 1 after injection; lasting at least three weeks). BoNT/A reduced pain symptoms only if injected after neuropathy onset, but not as a pretreatment.	[81]
paclitaxel-induced peripheral polyneuropathy in rats	abobotulinumtoxinA (AboA); 20-30 U/kg; i.pl; post-treatment	Antihyperalgesic effect at both ipsilateral and contralateral paws (three and six days after injection).	[82]
streptozotocin diabetic neuropahy in rats	A; 3, 5 and 7 U/kg (i.pl); 1 U/kg (i.t.); post-treatment	Unilateral toxin application reduced mechanical and thermal hypersensitivity bilaterally (from fifth to 15th day after BoNT/A). Intrathecal BoNT-A was effective after 24h. Different onset and lower analgesic dose after intrathecal injection suggested central action of BoNT/A.	[83]
chronic constriction injury to the sciatic nerve in mice and in rats	A;1.875, 3.75, 7.5 and 15 pg/paw for mice; 18. 75 or 75 pg/paw or i.t. for rats; post-treatment on day 5	Single i.pl. or i.t. injection significantly reduced the mechanical allodynia in mice and rats and thermal hyperalgesia in rats (from 24 h after toxin injection) and lasted for several weeks). Acceleration of regenerative processes in the injured nerve was also observed.	[84]
chronic constriction injury of the sciatic nerve in rats	A; 75 pg/paw; i.pl.; 3 days before and 5 days after CCI	Reduced neuropathic pain-related behavior and attenuated upregulation of NOS1, prodynorphin, pronociceptin mRNA in the DRG and microglia activation in both the spinal cord and DRG.	[85]
L5 ventral root transection (VRT) in rats	A; 7 U/kg; i.pl.; post-injury 4, 8 or 16 days	Reduced mechanical allodynia bilaterally and inhibited P2X (3) over-expression in DRG nociceptive neurons unulaterally to L5 VRT.	[46]
Infraorbital nerve constriction (IoNC) in rats	A; 3.5 U/kg into vibrissal pad; post-injury on day 14	Unilateral toxin injection reduced the IoNC-induced dural extravasation and allodynia bilaterally (from day 2 and lasting 17 days after BoNT/A, prior to neuropathy resolution). Intraganglionic block of axonal transport by colchicine abolished the effects of BoNT/A. Bilateral effects of BoNT/A and dependence on retrograde axonal transport suggest a central site of its action.	[86]
transection of the L5 ventral root in rats	A; 10 or 20 U/kg, i.pl. post-injury at day 3	Bilaterally decreased mechanical hyperalgesia, (from day 5, lasting at least 20 days post-BoNT/A). BoNT/A lowered the VRT-induced increased percentage of TRPV1 (+) neurons in the ipsilateral DRG.	[87]
chronic constriction injury in mice	A; 15 pg/paw i.pl.; post-injury at day 4	Counteracted allodynia and reduced astrocyte activation. It increased the analgesic effect of morphine and countered morphine-induced tolerance. In neurons BoNT/A restored the expression of MORs reduced by repeated morphine administration.	[88]
partial sciatic nerve transection in rats	A; 7 U/kg, i.pl. post-injury at day 14	Decreased mechanical and cold allodynia. Opioid antagonist naltrexone applied five days after the toxin reversed its antinociceptive effect. Central antinociceptive action of BoNT/A might be associated with the activity of endogenous opioid system via μ-opioid receptor.	[27]
partial sciatic nerve transection in rats	A; 7 U/kg, i.pl. post-injury at day 14	Reduced mechanical allodynia. GABA-A antagonist bicuculine abolished the antinociceptive effect in toxin-treated animals, thus indicating involvement of central GABAergic system.	[89]
chronic constriction injury of the infraorbital nerve in rats	A; 3 or 10 U/kg; s.c. into the whisker pad; post-injuryt at day 14	The toxin exerted antinociceptive effect and significantly lowered the expression of TRPA1, TRPV1, and TRPV2 in trigeminal nucleus caudalis (Vc); these effects were blocked by colchicine.	[90]
surgical constriction of the infraorbital nerve in rats	A; 15 U/kg; post-surgery at day 6; injected into the area of nerve ligation	Reduced thermal nociceptive response (TNR) beginning 6 h and lasting 72 h after treatment in senzitized animals. BoNT/A in sham group increased TNR thus suggesting a pronociceptive effect in non-sensitized animals.	[91]
malpositioned dental implants to induce injury to the inferior alveolar nerve in rats	A; 1 or 3 U/kg s.c. into the facial region; 3 days post-injury	Attenuated mechanical allodynia. Double treatments with 1 U/kg of BoNT-A produced prolonged, more antiallodynic effects as compared with single treatments. BoNT-A significantly inhibited the upregulation of Nav1.7 expression in the trigeminal ganglion in the nerve-injured animals.	[92]
chronic constriction injury of the sciatic nerve in rats	A; 300 pg/paw; i.pl. post-surgery at day 5	Attenuated pain-related behavior and microglial activation. It restored the neuroimmune balance by decreasing the levels of pronociceptive factors (IL-1β and IL-18) and increasing the levels of antinociceptive factors (IL-10 and IL-1RA) in the spinal cord and DRG.	[93]
streptozotocin-induced diabetic polyneuropathy; chronic constriction injury in rats	aboA; 15 or 20 U/kg; s.c.; post-injection and post-injury at day 14	Unilateral aboA reduced bilateral mechanical hyperalgesia in diabetic polyneuropathy model, while had no effect on unilateral CCI-induced hyperalgesia if applied contralaterally to the injury.	[94]
rat spared nerve injury (SNI) model	LC/E-BoNT/A chimera; 15–75 U/kg, i.pl. post-surgery at day 4	Alleviated for ∼two weeks mechanical and cold hyper-sensitivities. When injected five weeks after injury, LC/E-BoNT/A still reversed fully-established mechanical and cold hyper-sensitivity.	[18]
partial sciatic nerve ligation in mice (SP and NK1R knockout mice)	A; 0.2 and 0.4 U/paw, i.pl. post-surgery at day 7	Reduced hyperalgesia in wild type animals, but not in gene-deleted groups, suggesting the necessary involvement of SP-ergic system in the antinociceptive activity of BoNT/A.	[95]

**Table 3 toxins-11-00459-t003:** Randomized clinical trials of BoNT/A for neuropathic pain.

Pain Condition	Number of Participants	Dose and Delivery Route	Primary Outcome		Reference
posttraumatic neuralgia ^1^	29	5 U/site max. 200 U i.d.	pain rating 0–10	BoNT/A − 1.9 placebo − 0.3	[34]
posttraumatic neuralgia ^2^	46	5 U/site max. 300 U i.d.	pain rating 0–10	BoNT/A − 1.9 placebo − 0.6	[107]
postherpetic neuralgia	60	5 U/site max. 200 U s.c.	pain rating 0–10	BoNT/A − 4.5 lidocaine − 2.6 placebo − 2.9	[141]
postherpetic neuralgia	30	5 U/site max. 100 U s.c.	pain rating 0–10	BoNT/A − 4.6 placebo − 0.5	[142]
postherpetic neuralgia	117	2.5 U/site max. 200 U i.d.	pain rating 0–10	BoNT/A − 1.2 placebo − 1.2	[140]
trigeminal neuralgia	42	5 U/site, 75 U i.d. or s.m.	pain rating 0–10	BoNT/A − 6.05 placebo − 1.88	[143]
trigeminal neuralgia	20	5U/site, 100 U s.c.	pain rating (0–10) frequency of paroxysms/day	BoNT/A − 6.5 placebo − 0.3 BoNT/A − 32.7 placebo − 0.1	[144]
trigeminal neuralgia	36	50 U s.c.	pain rating (0–10) frequency of paroxysms/day	BoNT/A − 4.1 placebo − 1.25 BoNT/A − 22.0 placebo − 9.81	[145]
trigeminal neuralgia	80	20 sites 25 or 75U i.d. or s.m.	pain rating 0–10	BoNT/A 25U − 4.24 BoNT/A 75U − 5.4 placebo − 2.96	[146]
diabetic neuropathy	18	4U/site, 50U	pain rating 0–10	BoNT/A − 2.53 placebo − 0.53	[147]

^1^ 4 patients had postherpetic neuralgia; ^2^ or postsurgical; i.d. intradermal, s.c. subcutaneous, s.m. submucosal.

**Table 4 toxins-11-00459-t004:** Randomized clinical trials of BoNT/A for low back pain and osteoarthritic pain.

Pain Condition	Number of Participants	Dose and Delivery Route	Primary Outcome		Reference
low back pain	31	40 U/site 200 U i.m.	% of responders (50% reduction in pain rating)	BoNT/A 73.3% placebo 25%	[149]
refractory shoulder pain	36	100 U i.a.	pain rating 0–10	BoNT/A -2.4 placebo -0.8	[150]
refractory painful total knee arthroplasty	54	100 U i.a.	% of responders (2-point reduction in pain ratings)	BoNT/A 71% placebo 35%	[151]
knee osteoarthritis	176	200 U or 400 U i.a.	pain rating 0–10	BoNT/A 200 U -1.6 BoNT/A 400U -2.1 placebo -2.1	[152]
knee osteoarthritis	121	200 U i.a.	pain rating 0–10	BoNT/A -2.2 placebo -2.5	[153,154]

i.m. intramuscular, i.a. intraarticular.

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
