# Peer review of "Mechanisms of Botulinum Toxin Type A Action on Pain"

_toxins, 2019, doi:10.3390/toxins11080459_

Round 1
Reviewer 1 Report
In this comprehensive review, the mechanisms of action of botulinum toxin type A on pain are discussed.
Many studies are performed in rats and the scoring of the effects on pain often involve motor responses which may be affected by botulinum toxin. This may be discussed more extensively.
Also regarding the effects of botulinum toxin on migraine controversy exists. A placebo effect is suggested and in a recent trial (pijpers et al.) no additive effect of botulinum toxin on migraine could be detected. This should be discussed more clearly. Also in other RCTs there may be a placebo effect in botulinum toxin treated patients because of unblinding due to the motor effects of botulinum
The results of the study in ref 139 were not available to me.
Reviewer 2 Report
This review manuscript gave a comprehensive perspective on mechanisms of BoNT/A treatment on pain. Author summarized the origin of BoNT/A treatment on pain, the mode of action of BoNT/A on neurons and the possible BoNT/A effect on peripheral sensory nerves which involved in pain treatment. In this reviewer's view, this is a very good review paper which give readers informative and useful knowledge regarding to pain management utilizing BoNT/A. I recommend to publish this review without any modification.
Author Response
Thanks for your comments.